# Release of Endogenous Nutrients Drives the Transformation of Nitrogen and Phosphorous in the Shallow Plateau of Lake Jian in Southwestern China

Yang Zhang [ID], Fengqin Chang *, Xiaonan Zhang [ID], Donglin Li [ID], Qi Liu [ID], Fengwen Liu and Hucai Zhang *[ID]

Institute for Ecological Research and Pollution Control of Plateau Lakes, School of Ecology and Environmental Science, Yunnan University, Kunming 650504, China
* Correspondence: changfq@ynu.edu.cn (F.C.); zhanghc@ynu.edu.cn (H.Z.)

**Abstract:** Eutrophication remediation is an ongoing priority for protecting aquatic ecosystems, especially in plateau lakes with fragile ecologies and special tectonic environments. However, current strategies to control the phosphorus (P) and nitrogen (N) levels in eutrophication sites have been mainly guided by laboratory experiments or literature reviews without in-field analyses of the geochemical processes associated with the hydrological and eutrophic characteristics of lakes. This study analyzed the water quality parameters of 50 sites at Lake Jian in May 2019, a moderate eutrophication shallow plateau lake, based on dissolved/sedimentary nitrogen, phosphorous and organic matter, grain size, C/N ratios and stable isotope ratios of $\delta^{13}C$ or $\delta^{15}N$ in sediments. The results showed that the average total nitrogen (TN) and total phosphorus (TP) concentrations in the lake water were 0.57 mg/L and 0.071 mg/L, respectively. The TN and TP contents of surface sediment ranged from 2.15 to 9.55 g/kg and 0.76 to 1.74 g/kg, respectively. Stable isotope and grain source analysis indicated that N in sediments mainly existed in organic matter form and P mainly occurred as inorganic mineral adsorption. Endogenous pollution contributed to >20% of TN. Furthermore, our findings showed that phosphorus was the nutrient that limited eutrophication at Lake Jian, unlike other eutrophic shallow lakes. In contrast, the nutrient levels in the sediment and input streams belonged entirely to the N-limitation state. The distinctness in release intensity of N and P could modify the N/P limitation in the lake, which affects algae growth and nutrient control. These results demonstrated that reducing exogenous nutrients might not effectively mitigate lake eutrophication due to their endogenous recycling; thus, detailed nutrient monitoring is needed to preserve aquatic ecosystems.

**Keywords:** plateau lake; eutrophication; nutrient limitation transformation

## 1. Introduction

Eutrophication, i.e., the excessive richness of nutrients in a lake or other water body, is one of the principal ecological disasters during the evolution of lakes [1]. Overloaded N, P and other biogenic elements from exogenous pollution in a water body directly cause an abnormal increase in the primary productivity of aquatic ecosystems, leading to eutrophication [2]. Nutrient control of the eutrophicated lake is one of the remediation strategies based on the nitrogen and phosphorus limitation of algae growth [3]. Previous studies demonstrated that in severely affected eutrophicated lakes, the nutrient limitation theory might not work [3]. However, some scholars believe that it is necessary to control all nutrient elements [2]. In recent years, considering the costs and excessive amount of nitrogen or phosphorus causing algae growth, the single nutrient limitation strategy has been proposed as an attempt to remediate the eutrophication of lakes [4].

Lake sediments are important potential reservoirs of N, P and biogenic elements that accumulate in lake water through physical diffusion, convection, resuspension, and other geobiochemical procedures after discharge [5]. Their recycling can also affect lake

water quality, resulting in endogenous pollution. Meanwhile, long-term accumulation and degradation of organic matter (OM) produced by terrestrial materials, phytoplankton, and aquatic plant residues are also the main sources for the release of N and P [6,7]. Before effectively solving endogenous pollution, eutrophication might become more serious, even with exogenous pollution cut off [8].

Despite large amounts of investments, including >USD 10 billion annually and globally, to control N and P influx in lakes, the eutrophication problem is still spreading and growing, leading to recurrent cyanobacteria blooming even after remediation [3]. This phenomenon indicates that the current nutrient influx controls are insufficient. The main reason could be that we lack a full understanding of the internal cycling nutrients and geological or biogeochemical conditions of lakes. The previous eutrophication control paradigm was based mainly on laboratory or field experimental results, which tended to simulate the effects of external input without considering biogeochemical processes and geological background in the catchment of lakes [4,6,7]. Thus, more effort is needed to improve our understanding of the nutrient dynamics in lakes and the recharge procedures from catchment to achieve the desired eutrophication mitigation solutions, especially in plateau lakes with fragile ecological environments.

During the past decade, there has been a growing consensus that the concentration of nutrients in the upper sedimentary or water samples provides a basic general understanding of catchment development and the distribution of nutrients in a specific area because their concentrations do not account for the local geological types and other bio-geochemical controls [9]. Variations in grain size and composition of the sediment samples must be considered when documenting spatial variations in elemental concentrations. Samples rich in chemically reactive fine-grained (<63 μm) sediments are likely to contain higher concentrations than a sample dominated by sand, even if both originated from the natural rock unit or contaminated soils [10]. Moreover, isotopes are also useful tools for tracking the sources of nutrients and estimating their inner cycle processes [11]. To address the recycling issues and sedimentary variations in lakes, many researchers propose the use of multi-geochemical indexes, including C, N and P isotopes and grain size, as well as nutrient fractions, to determine the concentration of nutrients [12].

Lake Jian is an ecologically and environmentally protected region in northwest Yunnan that plays an important role in maintaining the regional ecological and environmental function and biodiversity [13,14]. Recently, developments in the catchment have drastically reduced the volume of Lake Jian, causing the lake to gradually wither [15]. Moreover, the nutrient element content has rapidly increased through agricultural production and other anthropogenic impacts, which have caused eutrophication and reduced the environmental function of lakes [16,17]. Most previous studies have focused on paleolimnology and paleoclimatology [18], ecological responses [13,16] and persistent environmental pollutants [19,20]. No comprehensive investigations have been conducted to reconstruct the reasons for eutrophication and estimate exogenous and endogenous pollution. Therefore, current nutrient levels, especially the levels of N and P in lake catchment systems, the source of pollutants and the internal reaction of biogenic elements, must be accurately and comprehensively explored.

Herein, 50 water and sediment samples from Lake Jian and its catchment rivers were analyzed to investigate the potential source and inner cycle of nutrient dynamics, specifically isotopes and grain size, in relation to lake trophic status and relative importance of N and P limitation.

## 2. Materials and Methods

### 2.1. Overview of Lake Jian

Lake Jian is a plateau fault lake located in the southern Hengduan Mountains in Dali prefecture, northwest Yunnan, Southwest China (Figure 1). It covers an area of 6.23 km$^2$, with an average water depth of 2.3 m. As one of the important upstream sources of Southeast Asian fluvial systems, more than five rivers originate from the basin but have

only one outlet, the Heihui River, which eventually joins the Yangbi River to become the principal branch of the Lantsang (Mekong) river and crosses six countries. Lake Jian has a maximum volume of 45.32 $\times$ 10$^6$ m$^3$, reflecting a decrease of more than 60% in recent years [14]. The mean air temperatures in winter and summer are 15 °C and 28 °C, respectively, and the mean annual rainfall is 786 mm.

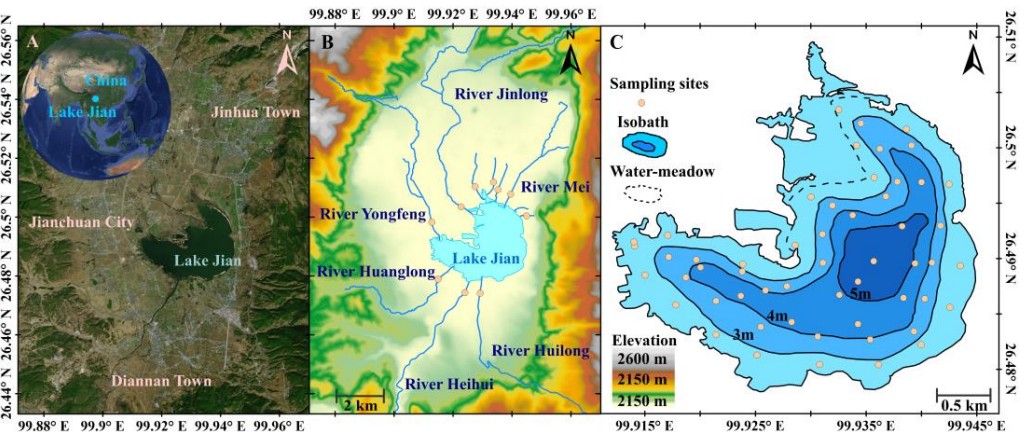

**Figure 1.** Location and geographic information (**A**), watershed system and digital elevation model of the catchment (**B**), lake isobaths (**C**) and sampling site ((**B**) rivers and streams; (**C**) lake water and sediment) of Lake Jian.

## 2.2. Sampling and Detecting

In May 2019, 50 surface sediment (each 0.5 cm thick), surface water and bottom water were sampled from Lake Jian, and 10 water samples from main streams and rivers into its catchment were collected using a gravity corer and condensate trap (Figure 1). All the samples were kept in brown polyethylene bottles and frozen in a refrigerator.

The nutrient element indices in water, including total nitrogen (TN) and total phosphorus (TP), were determined by the alkaline potassium persulfate digestion-UV spectrophotometric method (GB 11894-89) and ammonium molybdate spectrophotometric method (GB 11893-89) formulated by the State Environmental Protection Administration of China (SEPAC). The results were taken from the average values determined at three parallel times, and the measurement errors were less than 1%. Water quality parameters, including depth, temperature, chlorophyll a (Chl-a), pH, and dissolved oxygen (DO), were measured using 6600-V2 YSI during sampling. TN and TP in the surface sediment were measured by sulfuric acid digestion-Kjeldahl determination (GB 7173-87) and the Mo-Sb colorimetric method (GB 7852-87), following the standard of SEPAC with resulting measurement errors <5%. Samples for grain-size analysis were pre-treated using $H_2O_2$ and HCl to remove organic matter and carbonates. Grain size distribution between 0.02 μm and 2000 μm was measured using a Malvern Mastersizer 2000 laser grain-size analyzer, before the samples were deionized and dispersed by $Na(PO_3)_6$. Three types of bulk (<4 mm, 4–64 mm and >64 mm) size fractions were analyzed, and the analysis was focused on the fine-grained fraction (closely related to element enrichment in the surface soils). OM in the surface sediment was measured by the loss on ignition method. C/N, $\delta^{15}N$, and $\delta^{13}C$ in the surface sediment were detected with a Thermo MAT-253, with analytical errors of less than 0.2%.

## 2.3. Statistical Analysis

When considering the geochemical background and sources of nutrients, their relationship with grain size composition of samples were used for calculation [9]. Generally, higher concentrations of major elements in the earth's crust were observed in chemically reactive fine-grained (<64 μm) sediments, even in the samples obtained from natural and anthropogenic conditions [10]. Thus, the nutrients strongly correlated with the fine-grained fraction were chosen as exogenous sources to identify elements that could account for

the differences in grain size and composition [21]. The LTS robust regression model was defined by a logarithmic regression [22], and geochemical conditions in the catchment area and possible sources were estimated by the regression analyses in Lake Jian (details in Section 3.5). The N-, and P-limitations of lake eutrophication for algae growth complied with the formulation of the "Redfield ratio", e.g., N-limitation, N/P < 10; P-limitation, N/P > 20 [23].

For $\delta^{15}N$ and $\delta^{13}C$ analyses, $\delta$ notation was used to represent isotopic ratio differences between samples and standard materials. The formulas are expressed as follows:

$$\delta^{13}N = \frac{(Rsp_N - Rst_N)}{Rst_N} \times 1000‰ \tag{1}$$

$$\delta^{15}C_{org} = \frac{(Rsp_{C_{org}} - Rst_{C_{org}})}{Rst_{CC_{org}}} \times 1000‰ \tag{2}$$

Here, $\delta^{15}N$ and $\delta^{13}C$ represent the differences (‰) from the Vienna PDB standard and atmospheric $N_2$, and Rst and Rsp represent the stable isotope ratio of $\delta^{13}C$ or $\delta^{15}N$ in standard samples. Based on the mass conservation hybrid model and linear mixed model, the contribution of N and OM from different sources could be estimated as follows [24]:

$$\delta^{13}N = \sum_{x=1}^{n} W_x \times \delta^{13}N_x \tag{3}$$

$$\delta^{13}C_{org} = \sum_{x=1}^{n} W_x \times \delta^{13}C_{org} \tag{4}$$

Here, $\delta^{15}Nx$ and $\delta^{13}Corgx$ represent the corresponding isotope ratio detected in different end members of samples [25,26]. In this study, the main sources of the sedimentary OM were considered to be plankton (with typical stable compositions of C/N: 5~8, $\delta^{13}C$: −32~−23‰ and $\delta^{15}N$: 5~8‰), macrophytes (10~30, −27~−20‰, −15~20‰), soil (8~15, −32~−9‰, 2~5‰), terrestrial C3 plants (15~40, −32~−22‰, −6~5‰), terrestrial C4 plants (15~40, −16~−9‰, −6~5‰), and sewage (6.6~13, −26.7~−22.9‰, 7~25‰). The principal sources of sedimentary N were as follows: agricultural fertilizer (with $\delta^{15}N$ from −4~4‰), domestic sewage (10~20‰), soil erosion (3~8‰), terrestrial OM (with avg. of 2‰), and endogenous OM (6.5‰) [25]. The $W_x$ was the contribution probability of different pollutant sources, computed by an iterative calculation model, which is as follows:

$$W_x = \frac{\left[\left(\frac{100}{i}\right) + (n_S - 1)\right]!}{\left(\frac{100}{i}\right)!(n_S - 1)!} \tag{5}$$

Here, i represents the increment coefficient of the calculation model, and nS represents the number of N sources. Isotope sources were calculated using IsoSource [24]. Statistical analysis, one-way analysis of variance (ANOVA) and Pearson correlation (PC) were implemented using PAST v4.0 [27].

## 3. Results and Discussion

### 3.1. Nutrient Level and Water Quality of Lake Jian

The spatial distributions of mean TN (range: 0.05–0.99 mg/L, average: 0.57 mg/L) and TP (range: 0.003–0.173 mg/L, average: 0.071 mg/L) contents in surface water were as follows: western areas > central part > eastern areas (Figure 2). Moreover, the mean contents of TN and TP in the sites around the lakeside were higher than in the lake basin. Similarly, the contents of TN and TP in the bottom water of Lake Jian also showed broad variations, which ranged between 0.09 and 0.76 mg/L (0.29 mg/L) and 0.003 and 0.140 mg/L (0.087 mg/L). Generally, the highest TN and TP values were observed at

the western end of the lake near the entrance of the Yongfeng River, but the minimum values were detected in the central and eastern lake areas. Additionally, higher mean concentrations of TN and TP were observed at the surface water compared with bottom water (Table 1). Except for some sites in the western part of Lake Jian, the whole lake was in a state of middle eutrophication, belonging to Level III water quality (SEPAC standard). Significant differences in nutrient levels from the entering rivers and streams were observed. TN (2.81 mg/L) and TP (0.428 mg/L) levels in the Yongfeng River, which crosses a population center carrying domestic sewage, were significantly higher than in the Huilong River (TN: 1.93 mg/L, TP: 0.065 mg/L), Mei River (0.52 mg/L, 0.062 mg/L) and Jinlong River (1.58 mg/L, 0.016 mg/L) and streams entering the basin (0.14–0.51 mg/L, 0.015–0.049 mg/L), and traversing rural villages and farms in the catchment basin. High nutrient element levels were observed in the influx river, which was consistent with that accumulated in the lake area near an estuary, explaining the spatial discrepancy of nutrient elements in water to some extent. Nutrient elements of only one outlet, the Haiwei River, were 1.55 mg/L and 0.052 mg/L, respectively.

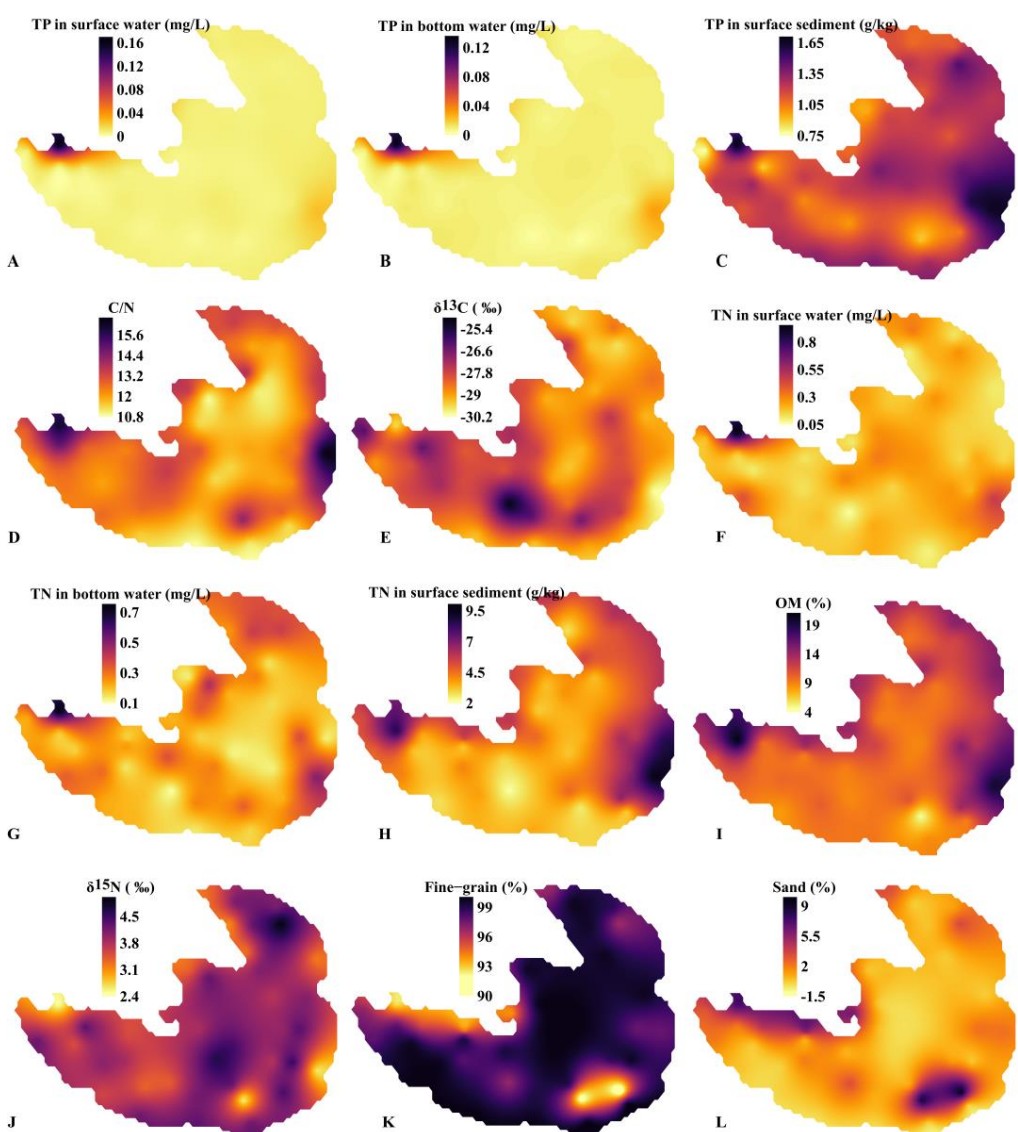

**Figure 2.** Environmental indexes of Lake Jian, including TN/TP in surface water (**A**,**F**), bottom water (**B**,**G**) and surface sediment (**C**,**H**), spatial distribution of C/N ratios (**D**), OM contents (**I**), $\delta^{13}$C (**E**), $\delta^{15}$N (**J**), fine-grain size (**K**) and sand fraction (**L**) composition.

**Table 1.** Nutrient element levels, water quality indices and geochemical information.

| Environmental Parameters | | Range | Environmental Parameters | | Range |
|---|---|---|---|---|---|
| TN | Surface water | 0.05~0.99 mg/L | TP | Surface water | 0.003~0.173 mg/L |
| | Bottom water | 0.09~0.76 mg/L | | Bottom water | 0.003~0.140 mg/L |
| | Surface sediment | 2.15~9.55 g/kg | | Surface sediment | 0.76~1.74 g/kg |
| | Output river | 1.55 mg/L | | Output river | 0.052 mg/L |
| | Rivers and streams | 0.14~2.81 mg/L | | Rivers and streams | 0.015~0.428 mg/L |
| | Pre-rivers and streams [a] | 0.04~1.85 mg/L | | Pre-rivers and streams [a] | 0.020~0.360 mg/L |
| | Pre-surface water [b] | 0.13~0.56 mg/L | | Pre-surface water [b] | 0.020~0.035 mg/L |
| Chl-a | Surface water | 16.90~63.68 μg/L | pH | Surface water | 8.34~9.23 |
| | Bottom water | 12.50~85.73 μg/L | | Bottom water | 8.21~9.17 |
| DO | Surface water | 6.53~10.96 mg/L | EC | Surface water | 250~284 μS/cm |
| | Bottom water | 1.57~8.87 mg/L | | Bottom water | 252~275 μS/cm |
| OM | Surface sediment | 4.25~21.30% | C/N | Surface sediment | 10.76~16.53 |
| $\delta^{15}N$ | Surface sediment | 2.45~5.00‰ | $\delta^{13}C$ | Surface sediment | −30.08~−24.08‰ |
| Clay | Surface sediment | 26.80~72.28% | Silt | Surface sediment | 27.49~67.38% |
| Fine-grain size | Surface sediment | 90.32~100% | Sand | Surface sediment | 0~9.68% |

Note: [a] [28]; [b] [16].

### 3.2. Spatial Distribution of Environmental Parameters in Surface Sediments

The spatial variability in TP, TN and OM concentrations in the surface sediment of Lake Jian is shown in Figure 2 (additional details in Table 1). OM concentrations in Lake Jian ranged from 4.25% to 21.30% (averaging 11.57%), with higher values observed on the western and eastern shores (>14.05%). There were lower contents of OM in the middle of the lake basin.

Generally, the distribution of TN in surface sediment is similar to that of OM, with a significant inward decreasing trend from the lakeshore (4.68–9.55 g/kg) to the deepest basin (2.15–4.03 g/kg). The mean concentration of TP decreased from the western lake margin (1.31 g/kg) to the middle (1.17 g/kg) and eastern parts of the lake (1.16 g/kg). In Lake Jian, the ratio of C/N was characterized by obvious spatial variation, with a range from 10.76 to 16.53, and an average value of 12.59. Higher C/N ratios were found in the sediment samples near Yongfeng River estuary (16.53) and the Mei River (15.03). Meanwhile, the C/N ratio was lower in the lake basin at its deepest portions (10.76~11.08), compared with other areas (11.17~13.05). The spatial distribution of $\delta^{13}C$ and $\delta^{15}N$ of surface sediment from Lake Jian is presented in Figure 2; their values ranged from −30.08 to −24.08‰ (averaging −28.28‰) and from 2.45 to 5.00 ‰ (3.84 ‰), respectively. $\Delta^{13}C$ exhibited a great spatial gradient, with its highest values recorded at the central lake bay area and lowest values recorded in the estuary of the western lake, while there was an overall decreasing trend from the eastern (−27.60‰) to the western part of the lake (−28.68‰). In contrast, $\delta^{15}N$ values showed regular spatial variation by isobath, with the most depleted $\delta^{15}N$ values occurring at the lakeshore (3.91–5.00‰), compared to the central basin (2.45–3.69‰).

The spatial variability in grain size fraction components is shown in Figure 2 (additional details in Table 1). To sort the particle size fractions, the percentage of clay components in the surface sediment samples of Lake Jian fell between 26.80 and 72.28%, with an average of 51.32%. There were obvious spatial differences between the central region and those close to the lake shore. The spatial change characteristics showed an increasing trend from the lake shore (49.46%) to the central region (56.73%). The silt component ranged between 27.50 and 67.38% (averaging 46.98%). The characteristics of spatial variation show that there was a decreasing trend from the lake shore to the lake center, i.e., the eastern lake area (54.78%) > the western lake area (48.32%) ≥ the central lake basin (41.75%). The percentage of sand components was between 0 and 9.68% (1.70%). Obvious high values were observed at the west lake bay near the entrance of the Jinlong River. Some high-value areas were also observed from the east bay to the central lake area. The mean contents of sand in Lake Jian were as follows: the western lake area (2.22%) > the central lake area (1.52%) > the eastern lake area (1.20%).

### 3.3. Source of Sedimentary OM and N

In this study, five OM sources (plankton, macrophytes, sewage, soil OM, terrestrial C3 and terrestrial C4) and N sources (agricultural fertilizer, domestic sewage, exogenous release, soil erosion, terrestrial input) were chosen to evaluate sources of pollutants (Figure 3).

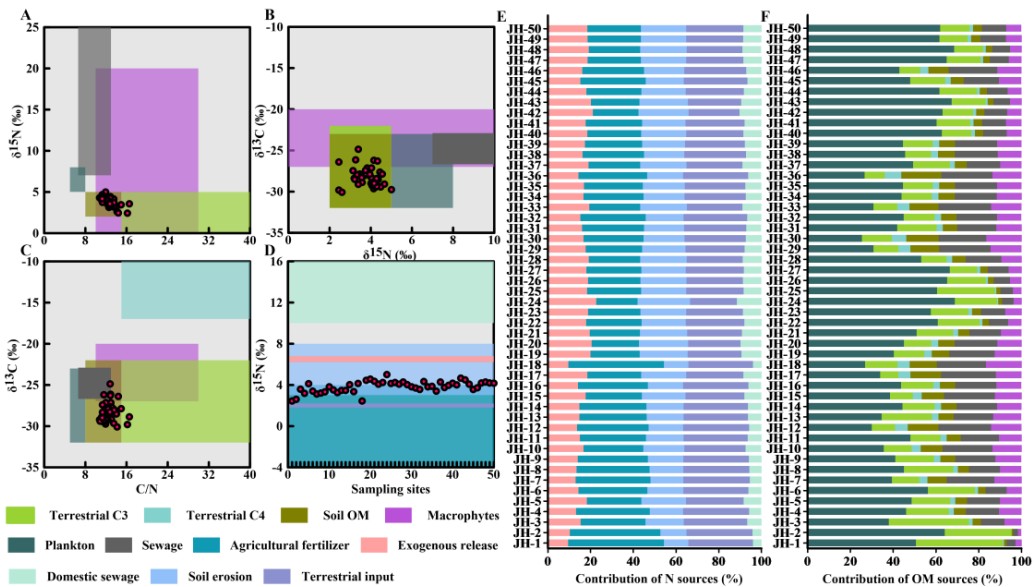

**Figure 3.** Scatterplots of $\delta^{15}$N/CN ratios (**A**), $\delta^{15}$N/$\delta^{13}$C (**B**) and $\delta^{13}$C/CN ratios (**C**), site-special range of $\delta^{15}$N (**D**) (isotopes boundary data from [25,26]), as well as the contribution of N (**E**) and organic matter sources (**F**).

Scatterplots of the $\delta^{15}$N and C/N (Figure 3A) in Lake Jian showed that sedimentary OM was mainly divided into two sources, macrophytes and soil OM. In addition, the results of $\delta^{13}$C vs. $\delta^{15}$N analyses (Figure 3B) showed that sedimentary OM sediments were primarily derived from terrestrial C3 plants and soil OM. The site-specific distribution of $\delta^{13}$C and C/N in Lake Jian was within the ranges of soil OM (Figure 3C). Based on the background range of source-specific environments (Figure 3D), $\delta^{15}$N in Lake Jian was mainly introduced from agricultural fertilizer and soil erosion.

In the increment coefficient of a calculation model for sources analysis, the mean and median values from the typically stable composition were used as the calculating $\delta^{13}$C values of plankton ($-27.5‰$), macrophytes ($-23.5‰$), sewage ($-24.8‰$), soil OM ($-27.5‰$), terrestrial C3 ($-27.0‰$) and terrestrial C4 ($-12.5‰$). Similarly, the C/N ratios of plankton, macrophytes, sewage, soil OM, terrestrial C3 and terrestrial C4 were 6.5, 20.0, 9.8, 11.5, 27.5 and 27.5, respectively [25]. Meanwhile, the $\delta^{15}$N values of plankton ($6.5‰$), macrophytes ($2.5‰$), sewage ($16.0‰$), soil OM ($3.5‰$), terrestrial C3 ($-0.5‰$) and terrestrial C4 ($-0.5‰$) were set as the source information into the IsoSource database [25,26]. For the multiple linear regression analysis in this study, an increase in the models was set as 1%, and the output data calculated were all reliable under the tolerance index of 3 ‰. The results shown in Figure 3 indicate that the contribution of plankton in sedimentary OM was 25.4–68.8% (average: 48.5%), 1.5–16.7% (9.7%) for macrophytes, 2.3–25.5% (15.7%) for sewage, 0.6–18.8% (6.7%) for soil OM, 8.5–40.7% (16.6%) for terrestrial C3 plant and 0.1–7.7% (2.8%) for terrestrial C4 plant. For the N sources calculation model, $\delta^{15}$N mean values of agricultural fertilizer, domestic sewage, exogenous release, soil erosion, and terrestrial input were 0.0‰, 15.0‰, 6.5‰, 5.5‰ and 2.0‰, respectively. The calculated mean contribution of agricultural fertilizer, domestic sewage, exogenous release, soil erosion, terrestrial input accounted for 28.4% (range: 19.3–44.8%), 7.5% (3.9–11.6%), 16.8% (9.6–22.7%), 19.3% (11.5–24.6%), and 28.0% (21.9–31.1%), respectively.

### 3.4. Exogenous Pollution due to N and P

The PC results showed that the correlation of TN (coefficient of correlation R = 0.70, *p* < 0.001) and TP (R = 0.99, *p* < 0.001) between surface water and bottom water samples in Lake Jian were significant (Figure 4A), indicating that the exchange and mixing of the upper and lower water body were sufficient. Considering a strong disturbance, the stable condition of solid nutrient elements in the surface sediment was destroyed, significantly exacerbating the migration and release of nutrient elements, especially in the shallow Lake Jian [29]. Higher dissolved nutrient element contents in the bottom water were found in Lake Jian, which was 1.2–1.4 times that in surface water (Figure 4C,D).

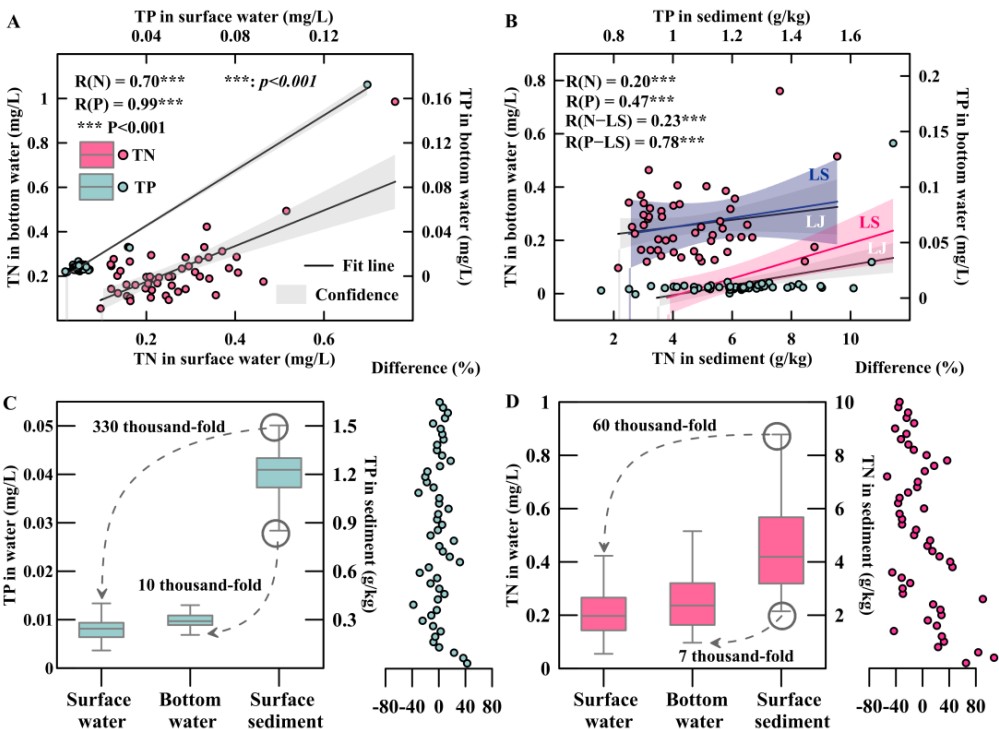

**Figure 4.** Relationship of TN and TP in surface water/bottom water (**A**) and bottom water/surface sediment (**B**), the contents levels of TP (**C**) and TN (**D**) in Lake Jian, with difference (%) in nutrient concentration between median value and site-specific data in surface sediment (**C,D**).

Except for the influence of deposition and gravity, vertical divergence in dissolved nutrient elements could be affected by the migration process near the sediment boundary by a series of bio-geochemical and geophysical processes [30,31]. The nutrient elements in the surface sediment were weakly correlated with TN (R = 0.20) and TP (R = 0.47) in the bottom water. All these results indicated that an increase in the concentrations of N and P in bottom sediments led to only a slight increase in the concentrations of these elements in waters and a slow release and migration from sediments into the lake water [32]. Recently, endogenous pollution of nutrient elements through release and migration has become one of the most important problems for pollution remediation and protection [33,34]. Previous studies have shown that exogenous pollution in lake systems was serious and difficult to control. In major eutrophic lakes worldwide, the endogenous load of nutrient elements exceeded 20–40% [35]. Noticeably, the TN (3.83–5.87 g/kg) and TP (1.16–1.31 g/kg) content in the surface sediment of Lake Jian were extremely high compared with other eutrophic lakes in China [36,37], indicating that endogenous pollution was likely to occur. The distribution difference between dissolved nutrient elements in water and sedimentary solid nutrient elements revealed that the release/migration process occurred at the sediment interface caused by endogenous pollution.

However, we found that the release intensity of N and P was remarkably different, and the fluctuation degree of TN ($4.85 \pm 1.02$ g/kg, with a maximum distance of 21.03%) content was much higher than that of TP ($1.24 \pm 0.07$, 6.07%) in surface sediment (Figure 4C,D). Under the same disturbed conditions and with similar input backgrounds, the results suggest that the deposition process of TP was more stable in Lake Jian. Furthermore, compared with nutrient element fluxes in the whole lake, the relative level of P content ($10.10$–$326.76 \times 10^3$ times the amount in the water body) in surface sediment was exceptionally higher than that of N ($7.72$~$59.17 \times 10^3$ times). Moreover, TP contents in the surface sediment and bottom water were more positively correlated (R = 0.44) compared with TN (0.24), especially in lakeside areas (P: 0.78, N: 0.23), indicating that the deposition process was relatively stable and there was no obvious endogenous release process (Figure 4B). On the other hand, the ANOVA result of N showed that the deposition state of solid N and dissolved N in lake water were two distinct processes ($p < 0.001$), and the mean relative level of N in surface sediment was nearly six times lower than that of P, indicating that the release of N could be more pronounced.

### 3.5. Endogenous Load and Different Forms of Nutrient Elements Pollutants

The pattern of endogenous pollution in a lake ecosystem is modulated by the distribution of distinct forms of nutrients and by environmental background limitations. Generally, under natural conditions, the different forms of N can be divided into organic nitrogen and inorganic nitrogen. Inorganic nitrogen mainly includes nitrate nitrogen, ammonia nitrogen, and nitrite nitrogen, introduced by agricultural activities, domestic sewage, industrial smelting, and other anthropogenic processes [38]. Additionally, through the absorption and transformation processes of microorganisms and aquatic animals and plants, organic nitrogen is abundant in the ecosystem, and could be in the form of biological debris and biological chain [39,40]. Both the degradation of organic nitrogen and the process of digestion/denitrification that involves inorganic nitrogen could lead to the migration and release of N in sediments [41]. However, under natural conditions, solid inorganic P in sediments produced by human activities were found to be mainly composed of ferric hydroxides, iron-manganese oxidizing material, and the AL–OM complex, which could reduce and decompose only in an anaerobic environment [42,43]. In comparison, the endogenous pollution of P was more limited than that of N, and irrespective of the effects of pH, temperature, visibility, dissolved oxygen, and other hydro-chemical environmental factors, or physical factors, such as biological disturbance intensity and wind/wave disturbance, it had more severe impacts on N release [43,44].

Nutrient elements and grain size could be influenced by additional factors, including provenance, degree of weathering, diagenesis, and biogenic production, requiring site-specific empirical data [45]. The main streams and their tributaries drain different catchments composed of variable bedrock geology (Figure 5C). Most river courses are predominantly of the Carboniferous system (gravelly coarse-grained and quartz sandstone), with minor basic metamorphics [21]. The upper reach of the Jinglong River was influenced by many tributaries draining Pleistocene, Jurassic, and Triassic sandstone and shales. The upper courses of the Huanglong and Huilong Rivers predominantly drain from Neogene and Quaternary formations. Overall, similar provenances in near-source areas did not influence the composition of minerals and phyllosilicates, which are typically the main carriers of trace and inorganic nutrient elements [21]. Areas of different provenance in our dataset did, nonetheless, display two opposing trends of N/FG and P/FG ratios, although we observed minor excursions from the general trend for FG, excluding the influence of samples from lake inlets and high-pollution outliers (Figure 5A,B). This result suggests that N concentrations in the surface sediment showed a relatively low sensitivity to provenance changes. Moreover, the occurrence form, combined with fine-grained minerals, was not the main speciation of sedimentary inorganic nitrogen contaminants. In contrast, high positive correlations of TP and FG in the surface sediment are observed in Lake Jian, indicating that the occurrence form combined with fine-grained minerals was the main speciation

of P, i.e., ferric hydroxides, iron–manganese oxidizing material inorganic fine-grained complexes [43].

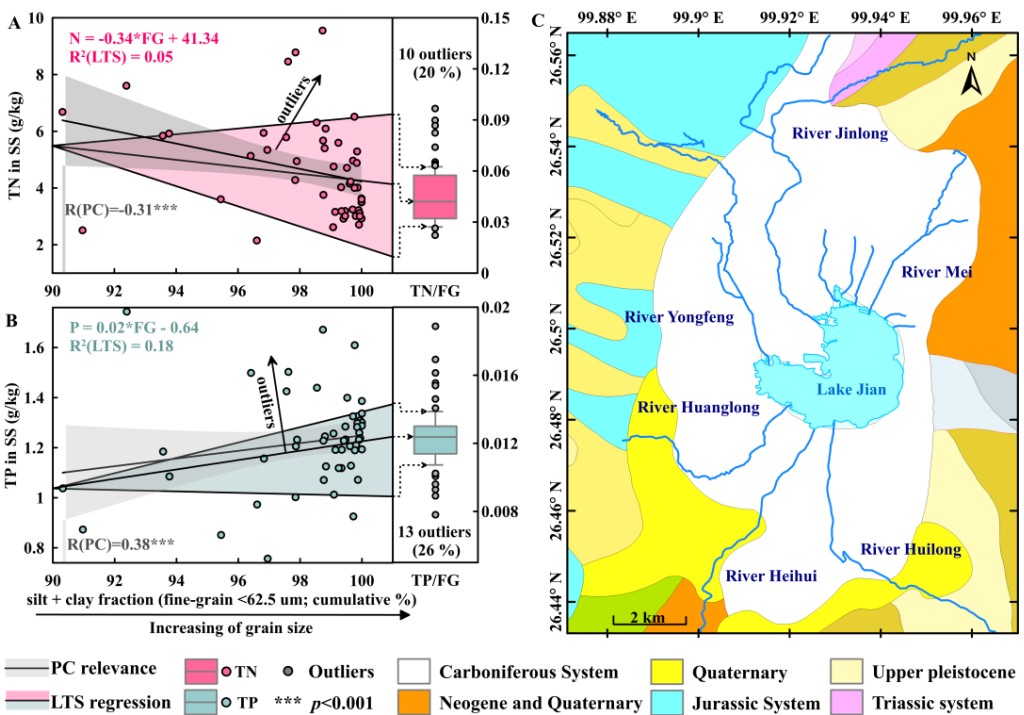

**Figure 5.** Bivariate plots of N (**A**) and P (**B**) vs. fine grain (FG), as well as Tukey's boxplots showing the geometrical form of absorbed mineral particle size based on the LTS robust regression, and geological stratigraphic map of the catchment of Lake Jian (**C**).

The results of biogenic element distribution showed that relative OM and TN levels in the surface sediment of Lake Jian were significantly higher than those of other lakes in Yunnan Province [25]. In addition to nutrient element levels, other factors, such as vegetation coverage and a well-developed river system, also increased TOC contents. Specifically, OM and TN contents in Lake Jian (R = 0.95) exhibited a significant linear correlation (Figure 6D), suggesting that more organic nitrogen was stored in Lake Jian. The situation was different with the PC results of TP and OM (0.36), which revealed that inorganic P was the main occurrence form of P in Lake Jian, which was then verified by traceability assessment. The OM in >75% of the sediment samples derived mainly from endogenous processes (plankton: 25.4–68.8%, averaging 52.53%; macrophytes: 3.7–16.6%, 8.99%), and only samples from entering river estuaries were related to domestic sewage (2.3–25.5% 17.47%), agricultural irrigation (soils OM: 0.6–14.4%, 7.37%), and crop planting (terrestrial C3: 8.5–40.7%, 18.35%; terrestrial C4: 0.1–6.0%, 3.1%). The source estimation of N showed that more than 16.8% of the sediment samples were derived from the sourcing process, which was remarkable for a lake with a low eutrophication level. Furthermore, we found that the level and distribution of N in surface sediment were less related to its material source (R−N/$\delta^{15}$N = 0.23) and occurrence form (R−N/$\delta^{13}$C = 0.28). ANOVA demonstrated that the sedimentary state of N was an independent process ($p < 0.001$). All this evidence indicates that the processes of N migration and release were determined by the current N level of the lake.

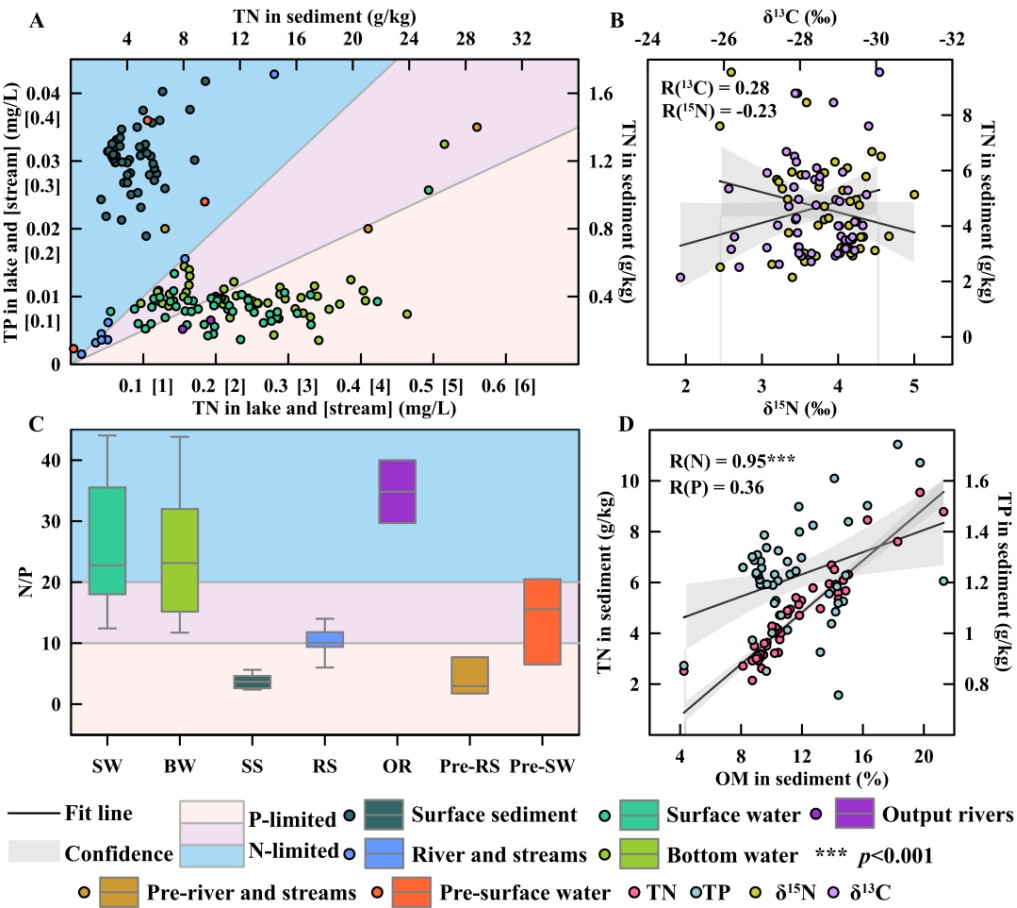

**Figure 6.** Ratio of TN and TP (**A**), correlation between TN and $\delta^{15}N/\delta^{13}C$ (**B**), TN/TP range (**C**), and correlation between nutrient elements in surface sediment and bottom water (**D**) in Lake Jian.

### 3.6. Transformation of N/P-Limitation Driven by Endogenous Pollution

Intriguingly, we found that the algal growth pattern of the Lake Jian water body was limited mainly by P (N/P ratios ranged from 5.7 to 64.2 times, averaging 26.5 times), while the nutrient elements levels in the sediment belonged entirely to the N-limitation state (2.1–7.3 times, 3.8 times). Generally, the strong exogenous pollution of N in the sediment, caused by its multiple unstable migrated forms and pathways, led to the enrichment of P [46,47]. These processes result in more dissolved N released into water, changing nutrient limitation. The hydrological results and lake monitoring records also support this conclusion (Figure 6A). There was also a great difference between the inflow of rivers and streams (5.8–13.2 times, 10.1 times) and the output river (29.5–45.3 times), meaning that whatever the river input via irrigation agriculture or city sewage, P was the dominating nutrient element and eutrophication was limited mainly by N. However, after the water entered Lake Jian, the exogenous pollution and dynamic processes changed the relative levels and structure of N/P, altering nutrient element limitation. Compared with previous monitoring data, the status of nutrient elements in Lake Jian has gradually changed from N-limitation (2000–2010) to P-limitation (2010–2018) in recent years [15,28,48]. This phenomenon was affected by the differences in cyclic processes and occurrences between N and P [49]. Due to the bio-accumulation of algae and aquatic plants, a large amount of external N in the lake is bounded by OM, enriching the sediments with biological death, completing the process of re-concentration of N in the lacustrine system [50]. Inevitably, combined with OM degradation and improved lake productivity, these processes re-intensify endogenous pollution and the enrichment condition of N, ultimately causing the transformation of nutrient element limitation.

We hypothesized that except for the influence of exogenous pollution, the primary cause of nutrient elements limiting their transformation in Lake Jian was their geophysical structure and environmental conditions in the catchment area. As a shallow lake, oxygen in the water-sediment interface in Lake Jian could be sufficiently supplied by air–water exchange and dynamic disturbance, alleviating anaerobic conditions by OM degradation [3]. Normally, the unstable states of P from anthropogenic activities that enter sediments by dynamic processes are principally excited as Al-Fe binding states, Ca-OM binding states, and residual P complexes [3,30]. Among them, hydrolysis and reduction of the Al-Fe-P complex, including ferric hydroxide and phosphoric iron, are the pathways for inorganic P release [46,49]. Under aerobic conditions, these materials are stable and immobile, inhibiting the migration and release of P.

A series of previous studies indicated that the external loss process of N (denitrification process) was strictly limited by oxygen content, and the loss process of P (dynamic deposition) was weakened, resulting in the formation of an N limitation state in shallow lakes, due to stronger water disturbance [3,37,49]. However, we observed dramatic OM levels and poor DO states at some sites, which were closely related to swamp formation processes at Lake Jian. In recent years, due to the impact of land reclamation and climate change, the lake area has decreased by more than 40% [14], resulting in most lakeside areas being exposed directly to the impact of the surrounding runoff. The original lakeside ecosystem has gradually evolved into a swampy area with dense vegetation. Despite high vegetation coverage having significantly alleviated the nutrient elements imported from external sources, it also played an important role in improving the transparency and oxygen content of the water body, limiting the occurrence of endogenous pollution processes, especially P release, which depend upon an anaerobic environment. However, due to the lack of exogenous pollution controls in early rising times, the huge algae masses produced by eutrophication and aquatic plants adsorbed innumerable nutrient elements in the lake sediments. Through the biological enrichment process, pollutants are enriched and transferred, especially the production of organic nitrogen, which could complete the migration and release process through various pathways, resulting in dissolved nitrogen frequently existing in the lake. Under an aerobic environment, the obstruction of gasification and efflux processes, such as denitrification, were also aggravated, finally achieving a P-limited state.

## 4. Conclusions

Based on the influence of environmental conditions, we observed several differences in the observed release/migration intensity of N and P in the sediments of Lake Jian, which resulted in the transformation of nutrients as limiting elements. We call this phenomenon the "pump diaphragm effect," which is the self-enrichment process of N, leading to the swamping of shallow plateau lakes. More remarkably, the water exchange cycle period was relatively long because of the closed environment and structural characteristics, which rendered the ecosystem of plateau lakes fragile and water self-restoration processes slow. Once the environmental background begins to fluctuate, the ecological and environmental health functions of plateau lakes could be irreversibly damaged. Our research results provide a sound theoretical basis and supporting data for improving the treatment of shallow, swamped plateau lakes. While decreasing/eliminating endogenous pollution, attention should also be focused on nutrient element circulation processes in the lake itself. More importantly, strategies that can control the nitrogen and phosphorus levels should be more carefully formulated due to complex and variable lake conditions; thus, the best strategy may be to implement the "one lake with one governance" strategy to fully understand the special characteristics of lakes and establish protocols for long-term eutrophication detection, assessment and management.

**Author Contributions:** Conceptualization, Y.Z. and H.Z; Formal analysis, D.L.; Funding acquisition, F.C. and H.Z.; Methodology, Y.Z.; Project administration, X.Z.; Resources, H.Z.; Software, Q.L. and F.L.; Supervision, F.C. and H.Z.; Writing—original draft, Y.Z.; Writing—review and editing, H.Z. All authors have read and agreed to the published version of the manuscript.

**Funding:** This research was funded by the Yunnan Provincial Government Scientist workshop and the Special Project for Social Development of Yunnan Province (Grant No. 202103AC100001, Funder: Hucai Zhang).

**Institutional Review Board Statement:** Not applicable.

**Informed Consent Statement:** Not applicable.

**Data Availability Statement:** Not applicable.

**Acknowledgments:** Special thanks are given to the team of the Key Laboratory of Plateau Lake Ecology and Global Change and the Institute for Ecological Research and Pollution Control of Plateau Lakes, who provided support with the sample collection and analysis.

**Conflicts of Interest:** The authors declare no conflict of interest.

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
