# Peer review of "Release of Endogenous Nutrients Drives the Transformation of Nitrogen and Phosphorous in the Shallow Plateau of Lake Jian in Southwestern China"

_water, doi:10.3390/w14172624_

Round 1

Reviewer 1 Report

Manuscript ID: water-1657879

Title: Endogenous pollutants drive the transformation of nitrogen and phosphorous limitation in a shallow plateau lake, Jianhu, southwestern China

Authors: Yang Zhang , Fengqin Chang * , Xiaonan Zhang , Donglin Li , Qi Liu , Fengwen Liu , Hucai Zhang *

This paper is well written, organized, and prepared. Conclusions are supported by sufficient experimental data and statistical analyses. The topic is well aligned with the Special Issues and is of great interest to the readers of Water. I had several minor questions for the authors’ consideration before publication.

Minor comments: 1. a better title may be needed, “Endogenous pollutants” is confusing to the readers. 2. The Abstract needs rewritting, the current form is not exciting. But the authors presented a nice and informative discussion in Section 4. 3. The Introduction is too short to give a strong background, for example, why the authors are interested in grain size of sediments? Any links to N and P release and cycle? 4. Some information is inconsistent or at least not clearly described, for example, the date in Abstract “May 2019” vs. in Section 2 “June 2018”; the data in the main text (e.g., Lines 111-112) vs. in Table 1. 5. Any references for Lines 127-129 and 146-147? 6. Please improve the quality and resolution of all figures. 7. Please explain the abbreviations, DEM in Figure 1 caption, Chl-a and DO in Line 78, EI and FG in Table 1, and abbreviations in plots (SW, BW, SS, etc.)

Author Response

Dear Reviewer,

We would like to express our sincere appreciation for your careful reading and invaluable comments to improve this manuscript. We have revised all issues raised by you. We believe the quality of this paper has been improved greatly by referring your positive opinion! Especially, the interesting and the readability of this paper has heightened significantly, after re-organization and major modification in section Abstract and Introduction. Besides, the description of methods have been included, the significance of granularity data has enhanced and cleared in this study. Moreover, the quality and resolution of all figures have improved. The line numbers in responses are based on the revised manuscript word version. More details about all the revisions are described in every response of your kindly comments.

Responses to Reviewer 1

Reviewer 1’s comments

This paper is well written, organized, and prepared. Conclusions are supported by sufficient experimental data and statistical analyses. The topic is well aligned with the Special Issues and is of great interest to the readers of Water. I had several minor questions for the authors’ consideration before publication.

Minor comments:

Point .1 a better title may be needed, “Endogenous pollutants” is confusing to the readers.

Response:

Thanks for your valuable suggestions. According to your suggestions, the title has been adjusted to “Release of endogenous nutrients drives the transformation of nitrogen and phosphorous in the shallow plateau of Lake Jian in southwestern China” . We believe this revision could avoid misleading for readers.

Point .2 The Abstract needs rewritting, the current form is not exciting. But the authors presented a nice and informative discussion in Section 4.

Response:

Thank you so much for your valuable suggestions. This is a very valid point. According to your suggestions, section Abstract has been revised. More details about our results and discussion have been added here. We also pointed out the impact of nutrient limitation transformation on the process of lake eutrophication and subsequent treatment . We believe these revisions can make the abstract more exciting and improve the wide interesting for international researches.

Point .3 The Introduction is too short to give a strong background, for example, why the authors are interested in grain size of sediments? Any links to N and P release and cycle?

Response:

Thanks for your valuable suggestions. According to your suggestions, the introduction section has been adjusted and modified. The links to N and P release processes have been added highlight the importance and scientific significance of nutrient cycling in eutrophication. Moreover, the N and P limitation, which affecting the algal growth, and how to determine it are also added in the section method. Scientific hypothesis plan and summary of previous researches are also added. The meaning of grain size and isotopes in sediments for identify the occurrence form of nutrients has been proposed here. Correspondingly, the description about the determination of grain size and reference analyses methods are also added in section method. We believe these revisions could emphasize key questions and research scientific significance in Lake Jian, and improve the scientificity and integrity of this research.

Point .4 Some information is inconsistent or at least not clearly described, for example, the date in Abstract “May 2019” vs. in Section 2 “June 2018”; the data in the main text (e.g., Lines 111-112) vs. in Table 1.

Response:

Thank you so much for the careful reminder. This is a important point. We has revised the inconsistent information in the section 2.2 and section 3.1. Moreover, we have revised and checked all the inconsistent information in section “results and discussion” .

Point .5 Any references for Lines 127-129 and 146-147?

Response:

Thank you so much for the careful reminder. This is a valid point. The data from section was detected by our lab work. Now, we add the description about the sampling and testing of rivers and stream samples in section 2.2. This revision can eliminate readers' doubts about the imperfect data acquisition.

Point .6 Please improve the quality and resolution of all figures.

Response:

Thank you so much for this advice. We has re-upload all the figures with high quality (at least 5000 Pixel Dimensions) and resolution (at least 500 dpi). In order to prevent possible compression of images during manuscript upload. We upload the PDF type and compressed package of all pictures this time.

Point .7 Please explain the abbreviations, DEM in Figure 1 caption, Chl-a and DO in Line 18, EI and FG in Table 1, and abbreviations in plots (SW, BW, SS, etc.)

Response:

Thank you so much for this advice. This is a valid point. We has revised the Figure 1 and the information about the location of the lake on the map of China and Province have been added now. Moreover, we changed the description about DEM to Digital elevation model in the Figure 1. Furthermore, The abbreviations about FG, SW, BW, SS and EI in Table1 and Figure 2 as well as Figure 5 have been deleted or changed to full spell. Abbreviations of Chl-a and DO in section 2.2 have also been explained.

Reviewer 2 Report

The manuscript is devoted to the important problem of studying the concentrations of nutrients (TN, TP) in surface and bottom waters, as well as bottom sediments in a shallow plateau lake, Jianhu, southwestern China. The results showed that endogenous pollutants drive the transformation of N and P limitation in this lake, which is a valuable result that can help in the study of the nutrient budget in other lakes. There are a number of issues.

Abstract.

Lines 14-15: It is indicated that the samples were taken in May 2019, but in the “Sampling and methods” section it is indicated that the samples were taken in June 2018 (lines 65-66). Please clarify.

Lines 22-24: Please, clarify.

Lines18-24: More attention needs to be paid to the results obtained in Sections 4.2 and 4.3.

  1. Sampling and methods

Line 63: “45.32×106 m3” – please use superscript (106).

Lines 68-69: This is a repeat about the average rainfall (lines 64-65).

Fig. 1: Please add the location of the lake on the map of China and Province. Decipher "DEM". Please add the location of the lake on the map of China and Province. Decipher "DEM".

Equation (4): There is no “δ13Corgx” in this equation (Why is nitrogen in the formula?).

Line 100: δ15N – please, use superscript.

Line 101: “soil erosion (3-8 soil erosion)” – please clarify “3-8”.

  1. Results

Lines 111-115: How were TN and NP concentrations measured in sediments so that the units would change to mg/L instead of g/kg (line 18)? Check the units.

Lines 125-129: The concentrations of TP and TN in the waters of the rivers flowing into the lake are given, but in the section 2 and in Fig. 1 it is not noted that samples were taken in the rivers too (only in the lake!).

Lines 130-133: It is not clear. Please rephrase and check if a part is missing in the second sentence.

Table 1: What is FG? How was clay, “slit” (silt?), and sand content determined?

Figure 2: There is no designation of A/F (surface water), B/G (bottom water), C/H (surface sediment) in the figure, specify the captions. δ15N is shown in fig. 2J, not in fig. 2F.

Section 3.3: There is no information about what methods and instruments were used to measure the grain size distribution and what classification of particle sizes was used.

Lines 117-186 and Fig. 3: Please provide references to data on the isotopic values to determine the likely sources of OM and N.

  1. Discussion and conclusion

Line 211: “R = 0.907”: is it the coefficient of correlation (R) or determination (R2)?

Lines 211-212: The R values in the text do not match the R values in Fig. 4A.

Lines 224-227: Low correlations between the N and P content in the bottom water and bottom sediments do not prove that their contents are “significantly influenced by release and migration”, since such correlations may, on the contrary, be due to the dynamic deposition process of dissolved nutrient elements, which contradicts the discussion on lines 224-227. So, according to Fig. 4B, an increase in the concentrations of N and P in bottom sediments leads only to a slight increase in the concentrations of these elements in waters, which may indicate the slow release and migration of these elements from sediments into waters. Please clarify.

Lines 245-256: “103 times” – please, use superscript (103).

Figure 4: What is shown in the figures “Distance in surface sediment (%)”, how was this index calculated?

Lines 239-253 and Fig. 4: How were fluxes of N and P calculated, what data on water reserves and bottom sediments in the lake were used? How did stock comparisons help estimate elements flow? It is necessary to describe the calculation methodology and pay more attention to the description and analysis of the results obtained in this subsection.

It is important to describe in more detail what results led to conclusions about the role of endogenous sources of N and P, as well as what the N-, and P-limitations indicate (and how they were established).

Lines 284-285: “N/FG and N/FG ratios” – please specify what ratios are calculated (now the same ones are indicated).

Lines 324-325: Specify units.

Grammar and spell checks are required.

Thus, the manuscript requires major revisions.

Author Response

Dear Reviewer,

We would like to express our sincere appreciation for your careful reading and invaluable comments to improve this manuscript. We have revised all issues raised by you. We believe the quality of this paper has been improved greatly by referring your positive opinion! Especially, the interesting and the readability of this paper has heightened significantly, after re-organization and major modification in section Abstract and Introduction. Besides, the description of methods have been included, the significance of granularity data has enhanced and cleared in this study. Moreover, the quality and resolution of all figures have improved. The line numbers in responses are based on the revised manuscript word version. More details about all the revisions are described in every response of your kindly comments.

Responses to Reviewer 2

Reviewer 2’s comments

The manuscript is devoted to the important problem of studying the concentrations of nutrients (TN, TP) in surface and bottom waters, as well as bottom sediments in a shallow plateau lake, Jianhu, southwestern China. The results showed that endogenous pollutants drive the transformation of N and P limitation in this lake, which is a valuable result that can help in the study of the nutrient budget in other lakes. There are a number of issues.

Point .1 Lines 14-15: It is indicated that the samples were taken in May 2019, but in the “Sampling and methods” section it is indicated that the samples were taken in June 2018 (lines 65-66). Please clarify.

Response:

Thank you so much for the careful reminder. This is a important point. We has revised the inconsistent information in the section 2.1.

Point .2 Lines 22-24: Please, clarify.

Response:

Thanks for your valuable suggestions. According to previous and your suggestions, the abstract section has been adjusted and modified. We clarified the missed and inconsistent information about N and P releasing result here.

Point .3 Lines18-24: More attention needs to be paid to the results obtained in Sections 4.2 and 4.3.

 Response:

Thanks for your valuable suggestion. This is a valid point. According to your suggestion, the description in abstract about the inconsistent results obtained in Sections 3.4 and 3.5 have been revised and checked again.

Point .4 Line 63: “45.32×106 m3” – please use superscript (106).

Response:

Thank you so much for your kind reminder. “45.32×106 m3” (line 103), the superscript has been used.

Point .5 Lines 68-69: This is a repeat about the average rainfall (lines 64-65).

Response:

Thank you so much for your kind reminder. The Repetitive sentence “ The mean air temperature in the lake basin is 12.3°C and mean annual rainfall is 786 mm.” has been deleted.

Point .6 Fig. 1: Please add the location of the lake on the map of China and Province. Decipher "DEM". Please add the location of the lake on the map of China and Province. Decipher "DEM".

Response:

Thank you so much for this advice. This is a valid point. We has revised the Figure 1 and the information about the location of the lake on the map of China and Province have been added now. Moreover, we changed the description about DEM to digital elevation model in the Figure 1.

Point .7  Equation (4): There is no “δ13Corgx” in this equation (Why is nitrogen in the formula?).

Response:

Thank you so much for your kind reminder. It was a wrong spell. We have been revised the Equation (4). “δ13Nx” has changed to “δ13Corgx” (line 156)

Point .8 Line 100: δ15N – please, use superscript.

Response:

Thank you so much for your kind reminder. “ δ15N”, the superscript has been used. We also checked all the superscript in whole MS.

Point .9 Line 101: “soil erosion (3-8 soil erosion)” – please clarify “3-8”.

 Response:

Thank you so much for your kind reminder. “soil erosion (3~8‰ soil erosion)”, the missing units has been used. We also checked all the units in whole MS. Moreover, “-” has revised to “~” in whole paper, in case the confusion to readers caused by minus.

Point .10 Lines 111-115: How were TN and NP concentrations measured in sediments so that the units would change to mg/L instead of g/kg (line 18)? Check the units.

Response:

Thank you so much for your kind reminder. This is a very valid point. We have revised and checked description about surface water and surface sediment and checked the units and range of TN and TP in samples in section 3.1. Moreover, wrong spelling of “aeras” has also checked and revised.

Point .11 Lines 125-129: The concentrations of TP and TN in the waters of the rivers flowing into the lake are given, but in the section 2 and in Fig. 1 it is not noted that samples were taken in the rivers too (only in the lake!).

Response:

Thank you so much for your kind reminder. The description about water samples collected from rivers and streams have been added to section 2 and Fig.1. Moreover, we also revised the section 1 and section 2 to explain why we interested in grain size of sediments and how to use it.

Point .12 Lines 130-133: It is not clear. Please rephrase and check if a part is missing in the second sentence.

Response:

Thank you so much for your kind reminder. We have revised and checked description here.

Point .13 Table 1: What is FG? How was clay, “slit” (silt?), and sand content determined?

Response:

Thank you so much for your kind reminder. We also checked the abbreviations about FG, SW, BW, SS and EI in Table1 and Figure 2 as well as Figure 5, which have been deleted or changed to full spell. Wrong spelling “slit” has been revised to “silt” now. Correspondingly, we also revised the section 1 and section 2 to explain why we interested in grain size of sediments and how to use it.

Point .14 Figure 2: There is no designation of A/F (surface water), B/G (bottom water), C/H (surface sediment) in the figure, specify the captions. δ15N is shown in fig. 2J, not in fig. 2F.

Response:

Thank you so much for your kind reminder. We have been revised and checked Figure 2. Now, the captions of all the sub-picture is consistent with the information.

Point .15 Section 3.3: There is no information about what methods and instruments were used to measure the grain size distribution and what classification of particle sizes was used.

Response:

Thank you so much for your kind reminder. This is a very valid point like Point 13. It is very necessary to tell readers how do we determine the content of each grain fraction and what classification of particle sizes was used. We have revised the section method. The description about the determination and classification of grain size, and reference analyses methods have been added.

Point .16 Lines 117-186 and Fig. 3: Please provide references to data on the isotopic values to determine the likely sources of OM and N.

Response:

Thank you so much for your kind reminder. References to data on the isotopic values to determine the likely sources of OM and N have been added now.

Point .17 Line 211: “R = 0.907”: is it the coefficient of correlation (R) or determination (R2)?

Response:

Thank you so much for your kind reminder. We have noted the coefficient of correlation (R) in the sentence.

Point .18 Lines 211-212: The R values in the text do not match the R values in Fig. 4A.

Response:

Thank you so much for your kind reminder. We have revised the inconsistent description here.

Point .19 Lines 224-227: Low correlations between the N and P content in the bottom water and bottom sediments do not prove that their contents are “significantly influenced by release and migration”, since such correlations may, on the contrary, be due to the dynamic deposition process of dissolved nutrient elements, which contradicts the discussion on lines 224-227. So, according to Fig. 4B, an increase in the concentrations of N and P in bottom sediments leads only to a slight increase in the concentrations of these elements in waters, which may indicate the slow release and migration of these elements from sediments into waters. Please clarify.

Response:

Thank you so much for your kind advice. This is a very good point. According to your advice. We have been revised the description about the results here.

Point .20 Lines 245-256: “103 times” – please, use superscript (103).

Response:

Thank you so much for your kind reminder. We have used superscript here.

Point .21 Figure 4: What is shown in the figures “Distance in surface sediment (%)”, how was this index calculated?

Response:

Thank you so much for your kind suggestion. We described more details on how to calculate distance in Figure 4.

Point .22 Lines 239-253 and Fig. 4: How were fluxes of N and P calculated, what data on water reserves and bottom sediments in the lake were used? How did stock comparisons help estimate elements flow? It is necessary to describe the calculation methodology and pay more attention to the description and analysis of the results obtained in this subsection.

Response:

Thank you so much for your kind reminder. It was not the fluxes of N and P but the times of N, P contents in water and sediment. We have revised the description here.

Point .23 It is important to describe in more detail what results led to conclusions about the role of endogenous sources of N and P, as well as what the N-, and P-limitations indicate (and how they were established).

Response:

Thank you so much for your kind reminder. This is a very valid point. We have been revised the MS, and give more evidence about endogenous sources of N and P in section 3.6. Moreover, the -, and P-limitations indication and establishing are also added in section Introduce and Method.

Point .24 Lines 284-285: “N/FG and N/FG ratios” – please specify what ratios are calculated (now the same ones are indicated).

Response:

Thank you so much for your kind reminder. We have been revised here.

Point .25 Lines 324-325: Specify units.

Response:

Thank you so much for your kind reminder. We added the been units here.

Reviewer 3 Report

This manuscript is a detailed study that provides interesting information on external loads and recycling of N and P in a plateau lake, as well as on their role on the eutrophication of this freshwater ecosystem. In general, the manuscript is properly written, the data are analyzed correctly and the figures are informative. However, I suggest a MINOR REVISION to improve the study in the following general and specific aspects.

General comments:

The structure of the study might be improved, dividing its content in “Results and Discussion” and “Conclusions”, rather than in the current sections (“Results” and “Discussion and Conclusions”). It should be noticed that the Figures 4,5,6 and the relevant statistical analyses are basically “results” of this study and not merely a discussion. Moreover, the absence of a short section at the end of the manuscript with the conclusions makes more difficult to identify the main results of this study by the readers.

The resolution of the Figure 3 should be increased, as titles and legend of this figure are not easily readable both in printed and electronic versions of the manuscript. The authors might also divide this figure in two parts, in order to increase the sizes of both panels.

Specific comments:

There is a number of specific points that might be improved. Please, check if the following suggestions can be appropriate.

Lines 19-20: … mainly exists in organic form ….

Lines 21-22: Please check if this is the meaning of this sentence: “The statistical analysis suggested that distinct releases of N and P from the sediments modify N/P limitation in lake environment”.

Lines 22-24: “These results demonstrate that the reduction of exogenous P or N might not effectively mitigate lake eutrophication, due to their endogenous recycling, thus a detailed nutrient monitoring is needed to preserve this aquatic ecosystem.”

Line 34: However, their recycling can also affect the water quality, ….

Line 132: … . Nutrient elements ….

Lines 252-253: … indicating that the release of N may be more pronounced.

Lines 255-257: This sentence is not very clear, perhaps the meaning is: “The pattern of endogenous pollution in lake ecosystem is modulated by the distribution of distinct forms of the nutrients and by environmental background limitations.”

Line 258: … . Inorganic nitrogen …

Author Response

Dear Reviewer,

We would like to express our sincere appreciation for your careful reading and invaluable comments to improve this manuscript. We have revised all issues raised by you. We believe the quality of this paper has been improved greatly by referring your positive opinion! Especially, the interesting and the readability of this paper has heightened significantly, after re-organization and major modification in section Abstract and Introduction. Besides, the description of methods have been included, the significance of granularity data has enhanced and cleared in this study. Moreover, the quality and resolution of all figures have improved. The line numbers in responses are based on the revised manuscript word version. More details about all the revisions are described in every response of your kindly comments.

Responses to Reviewer 3

Reviewer 3’s comments

This manuscript is a detailed study that provides interesting information on external loads and recycling of N and P in a plateau lake, as well as on their role on the eutrophication of this freshwater ecosystem. In general, the manuscript is properly written, the data are analyzed correctly and the figures are informative. However, I suggest a MINOR REVISION to improve the study in the following general and specific aspects.

General comments:

Point .1 The structure of the study might be improved, dividing its content in “Results and Discussion” and “Conclusions”, rather than in the current sections (“Results” and “Discussion and Conclusions”). It should be noticed that the Figures 4,5,6 and the relevant statistical analyses are basically “results” of this study and not merely a discussion. Moreover, the absence of a short section at the end of the manuscript with the conclusions makes more difficult to identify the main results of this study by the readers.

Response:

Thank you so much for your kind suggestion. This is a very great point. We have revised the section Results, Discussion and Conclusions. Now, section 3. Results, Discussion and section 4 Conclusions are being a single part. We believe these revisions can make the section Discussion and Conclusions more clear and improve the wide interesting for readers.

Point .2 The resolution of the Figure 3 should be increased, as titles and legend of this figure are not easily readable both in printed and electronic versions of the manuscript. The authors might also divide this figure in two parts, in order to increase the sizes of both panels.

Response:

Thank you so much for your kind suggestion. Including the revised Figure 3 with more details, we has re-upload all the figures with high quality (at least 5000 Pixel Dimensions) and resolution (at least 500 dpi). In order to prevent possible compression of images during manuscript upload. We upload the PDF type and compressed package of all pictures this time.

Specific comments:

There is a number of specific points that might be improved. Please, check if the following suggestions can be appropriate.

Point .3 Lines 19-20: … mainly exists in organic form ….

Response:

Thank you so much for your kind reminder. We have revised the description here.

Point .4 Lines 21-22: Please check if this is the meaning of this sentence: “The statistical analysis suggested that distinct releases of N and P from the sediments modify N/P limitation in lake environment”.

Response:

Thank you so much for your kind suggestion. We are very agree with this point, and the sentence has revised here.

Point .5 Lines 22-24: “These results demonstrate that the reduction of exogenous P or N might not effectively mitigate lake eutrophication, due to their endogenous recycling, thus a detailed nutrient monitoring is needed to preserve this aquatic ecosystem.”

Response:

Thank you so much for your kind suggestion. We are very agree with this point, and the sentence has revised here.

Point .6 Line 34: However, their recycling can also affect the water quality, ….

Response:

Thank you so much for your kind reminder. We have revised the description here.

Point .7 Line 132: … . Nutrient elements ….

Response:

Thank you so much for your kind reminder. We have revised the capital letter in the top of the sentence here.

Point .8 Lines 252-253: … indicating that the release of N may be more pronounced.

Response:

Thank you so much for your kind reminder. We have revised the description here.

Point .9 Lines 255-257: This sentence is not very clear, perhaps the meaning is: “The pattern of endogenous pollution in lake ecosystem is modulated by the distribution of distinct forms of the nutrients and by environmental background limitations.”

Response:

Thank you so much for your kind suggestion. We are very agree with this point, and the sentence has revised here.

Point .10 Line 258: … . Inorganic nitrogen …

Response:

Thank you so much for your kind reminder. We have revised the description here.

Round 2

Reviewer 2 Report

The authors fully responded to my comments and made the required corrections to the text. Therefore, the manuscript can be accepted for publication in its current form.